# Multiomics Reveals the Effect of Root Rot on Polygonati Rhizome and Identifies Pathogens and Biocontrol Strain

Zhiqiang Pang,[a,b,c] Xinyu Mao,[a,b] Yong Xia,[d] Jinxian Xiao,[e] Xiaoning Wang,[f,g] Peng Xu,[a,b,c] Guizhou Liu[a]

aCrops Conservation and Breeding Base, CAS Key Laboratory of Tropical Plant Resources and Sustainable Use, Xishuangbanna Tropical Botanical Garden, Chinese Academy of Sciences, Menglun, China
bCollege of Life Sciences, University of Chinese Academy of Sciences, Beijing, China
cThe Innovative Academy of Seed Design, Chinese Academy of Sciences, Menglun, China
dInstitute of Geochemistry, Chinese Academy of Sciences, Guiyang, China
eSchool of Biological and Chemical Science, Pu'er University, Puer, China
fKey Laboratory for Crop Breeding of Hainan Province, Haikou, China
gSanya Institute, Hainan Academy of Agricultural Sciences, Sanya, China

Zhiqiang Pang and Xinyu Mao contributed equally to this article. Author order was determined on the basis of seniority.

**ABSTRACT** Root (rhizome) rot of *Polygonatum* plants has received substantial attention because it threatens yield and sustainable utilization in the polygonati rhizome industry. However, the potential pathogens that cause rhizome rot as well as the direct and indirect (via root-associated microbes) strategies by which *Polygonatum* defends against pathogens remain largely unknown. Herein, we used integrated multiomics of plant-targeted metabolomics and transcriptomics, microbiome, and culture-based methods to systematically investigate the interactions between the *Polygonatum cyrtonema* Hua root-associated microbiota and pathogens. We found that root rot inhibited *P. cyrtonema* rhizome growth and that the fresh weight significantly decreased ($P < 0.001$). The transcriptomic and metabonomic results showed that the expression of differentially expressed genes (DEGs) related to specialized metabolic and systemic resistance pathways, such as glycolysis/gluconeogenesis and flavonoid biosynthesis, cycloartenol synthase activity (related to saponin synthesis), mitogen-activated protein kinase (MAPK) signaling, and plant hormone signal transduction, was particularly increased in diseased rhizomes. Consistently, the contents of lactose, D-fructose, sarsasapogenin, asperulosidic acid, botulin, myricadoil, and other saponins, which are functional medicinal compounds present in *P. cyrtonema* rhizomes, were also increased in diseased plants infected with rhizome rot. The microbiome sequencing and culture results showed that root rot disrupted the *P. cyrtonema* bacterial and fungal communities and reduced the microbial diversity in the rhizomes and rhizosphere soil. We further found that a clear enrichment of *Streptomyces violascens* XTBG45 (HJB-XTBG45) in the healthy rhizosphere could control the root rot caused by *Fusarium oxysporum* and *Colletotrichum spaethianum*. Taken together, our results indicate that *P. cyrtonema* can modulate the plant immune system and metabolic processes and enrich beneficial root microbiota to defend against pathogens.

**IMPORTANCE** Root (rhizome or tuber) reproduction is the main method for the agricultural cultivation of many important cash crops, and infected crop plants rot, exhibit retarded growth, and experience yield losses. While many studies have investigated medicinal plants and their functional medicinal compounds, the occurrence of root (rhizome) rot of plant and soil microbiota has received little attention. Therefore, we used integrated multiomics and culture-based methods to systematically study rhizome rot on the famous Chinese medicine *Polygonatum cyrtonema*

Address correspondence to Guizhou Liu, liugz@xtbg.ac.cn, or Peng Xu, xupeng@xtbg.ac.cn.

The authors declare no conflict of interest.

and identify pathogens and beneficial microbiota of rhizome rot. Rhizome rot disrupted the *Polygonatum*-associated microbiota and reduced microbial diversity, and rhizome transcription and metabolic processes significantly changed. Our work provides evidence that rhizome rot not only changes rhizome transcription and functional metabolite contents but also impacts the microbial community diversity, assembly, and function of the rhizome and rhizosphere. This study provides a new friendly strategy for medicinal plant breeding and agricultural utilization.

**KEYWORDS** medicinal plant, rhizome rot, root microbiota, PSM, polysaccharide and saponins, *Streptomyces*

*P*olygonatum cyrtonema Hua ("Duohua Huangjin" in Chinese) is a known Chinese herbal medicine, and its rhizome (*Polygonatum* rhizoma) is used as food and medicine to prevent and treat various diseases, such as fatigue, dizziness, cough, and other respiratory problems, and to improve memory and enhance immunity (1). Previous studies have shown that the functional medicinal ingredients of *P. cyrtonema* are attributed to its steroidal and triterpenoid saponins, polysaccharides, flavonoids, and other components (2, 3). These components possess functional bioactivities, including antioxidative, antidiabetic, antitumor, anti-inflammatory, antiviral, immunomodulatory, and neuroprotective activities (4). Therefore, the *P. cyrtonema* rhizome and its functional components have good utilization prospects in the development of useful medicine and food. Many health care products and medicines containing *Polygonatum* rhizome have been rapidly developed, and the annual demand for processed *Polygonatum* rhizomes is approximately 4,000 tons in China (5). However, rhizome reproduction is the main method for the agricultural cultivation of *Polygonatum* rhizomes, and infected *P. cyrtonema* plants display blight and yellow leaves, retarded growth, root rot (rhizome and stem bases), and yield losses. Root rot is a large factor constraining rhizome yield and *Polygonatum* marketability in China (6).

Some studies have characterized the root rot of various tuberous or rhizome plants, such as potato (7), Banxia (*Pinellia ternata*) (8), Aconitum (9), turmeric plants (10), lotus (11), Sanqi (*Panax notoginseng*) (12, 13), and ginger (14, 15), and pathogenic fungi were identified. The changes in the microbial community (16) and the defense response of plants were also studied (17, 18). Previous studies have shown that plant metabolites and microbiota contribute to host plant defense against pathogens (19–21). For example, previous studies have summarized the important role of polysaccharide (22, 23) and steroidal and triterpenoid saponin toxins (24, 25) in disease resistance. Similarly, there is increasing evidence that the root/rhizosphere microbiota directly or indirectly participate in host plant metabolic processes by regulating host secondary or special metabolism (19, 26, 27). Multiomics approaches are comprised of metabolomics, transcriptomics, phenomics, and microbiome integrated studies on the interaction of host plants with the microbiota and their external environment and generate more layered information that can explain the mechanisms involved in root rot (28–30). To ensure more effective and stable biocontrol of crop disease in the field, it is necessary to have a clear understanding of the interaction between the soil microbiota community, biocontrol microbes, and pathogens (17, 31, 32). Using a microbial strategy (microbiome) to control diseases has been successful in many ways (9, 33–35), and biocontrol is considered to be a desirable approach for controlling host plant diseases or soilborne pathogens (7, 36). Moreover, because of the large genome and long breeding cycle of resistant *P. cyrtonema* varieties, it is feasible to use microbial methods to control diseases and identify potential beneficial microbes. In addition, no studies have systematically studied the effect of root rot on *P. cyrtonema* gene transcription, functional metabolites, or belowground microbiota.

It is of great significance to conduct multiomics studies on plant diseases. Multiomics research on healthy and diseased rhizomes can further identify pathogens and beneficial microbes and reveal the defense response of host plants and functional microbiota against

pathogens. We hypothesized that diseased rhizomes induced changes in *P. cyrtonema* gene transcription, metabolic processes, and belowground bacterial and fungal communities. To test these hypotheses, we investigated the transcriptome, metabolome, and microbiome (rhizome, rhizosphere, zone and furrow soil) of *P. cyrtonema*. The objectives of this study were as follows: (i) elucidate the effect of root rot on the functional medicinal components and gene transcription of the *P. cyrtonema* rhizome and determine which functional metabolites and genes are involved in the disease resistance of *P. cyrtonema*, (ii) determine the range of the microbial community characteristics associated with diseased rhizomes, and (iii) identify potential pathogens and functional microbes through the microbiome. Ultimately, revealing the features of rhizome defense and root-associated beneficial microbiota in resistant *P. cyrtonema* should provide new ideas for breeding and agricultural utilization as well as increased tolerance and even resistance to rhizome rot disease.

## RESULTS

**Root rot changes *P. cyrtonema* rhizome metabolism and reduces the fresh weight.** The diseased *P. cyrtonema* displayed leaves and stems that were withered and yellow, rhizome rot, retarded growth, and plant death, and the yield was reduced (Fig. 1A and Fig. S1). It was evident that root rot inhibited rhizome growth, and the fresh weight was significantly decreased ($n = 6$; average, 257.22 g) compared with that of healthy plants ($n = 6$; average, 444.27 g; $P < 0.001$) (Fig. 1C; Table S1 in the supplemental material). To systematically evaluate the effects of rhizome rot on the growth and medicinal functional components of rhizomes, we assessed the targeted metabolite contents (13 polysaccharides and 44 saponins) and transcriptomes of healthy and diseased rhizomes (Table S1). Surprisingly, some of the *P. cyrtonema* rhizome medicinal components had a positive response to root rot. After rhizome rot, the contents of lactose and D-fructose in the diseased rhizome ($n = 12$; average of 0.40 and 10.45 mg g$^{-1}$) were higher than those in healthy plants ($n = 12$; 0 and 0.58 mg g$^{-1}$, average, respectively; Fig. 1C). The contents of polygonatone B/C, diosgenin, sarsasapogenin, asperulosidic acid, botulin, myricadoil, and others were increased compared with those in the healthy plants (Fig. 1B and E). However, the contents of geniposide, oleanolic acid, 24,30-dihydroxy-12(13)-enolupinol, ursolic acid, betulinic acid, mangiferolic acid, and others changed only slightly in the diseased rhizome compared with the healthy plant rhizome (Fig. 1D; Table S1). These findings indicate that *P. cyrtonema* polysaccharides and saponins, particularly steroidal and triterpene saponins, triterpenes, and monoterpenoids, might be involved in the defense of plants against rhizome rot, and polysaccharides signaling may also contribute to immune responses against pathogens.

**Transcriptome studies confirm the changes in metabolic processes and the defense system of *P. cyrtonema* rhizomes.** To further elucidate the above metabolic processes underlying changes in rhizome metabolites at the molecular level, we conducted comparative transcriptomics analyses on six diseased and six healthy samples to assess the global rhizome response to root rot. According to transcriptome and metabolome association analyses, we concluded that similar changes in transcription and metabolism occurred in the diseased plants as in the healthy plants. In addition, consistent with the metabolite analysis, a significant number of metabolic pathways were found to be related to polysaccharides (starch, sucrose, fructose, mannose, and galactose metabolism), saponins, and other metabolites, which targeted *P. cyrtonema* rhizome medicinal components (Fig. S2 and Table S2). First, a principal-component analysis (PCA) revealed that the healthy and diseased plants were clustered (Fig. S2). It was evident that there were differences in gene expression in diseased plants compared with healthy plants. The transcriptome comparative analysis resulted in the identification of a total of 28,221 differentially expressed genes (DEGs; 23,423 upregulated genes and 4,798 downregulated genes) that showed differential expression patterns in the diseased rhizome compared with the healthy rhizome (Fig. 2A; Table S2). To identify metabolic pathways and terms that were potentially associated with the root rot resistance of *P. cyrtonema*, we carried out a standard pathway enrichment analysis based on Gene Ontology (GO) and Kyoto Encyclopedia of Genes and Genomes (KEGG).

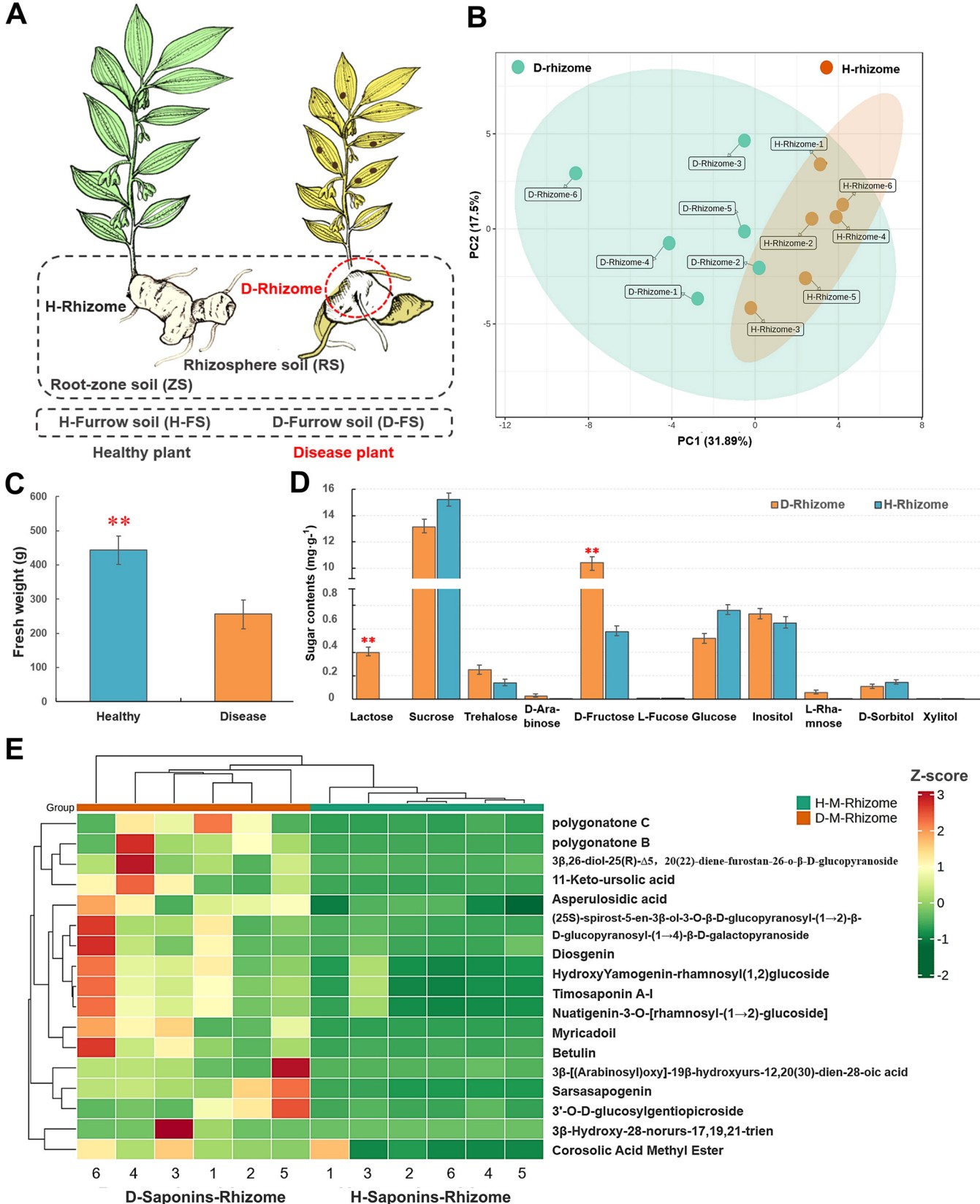

**FIG 1** Rhizome metabolite analysis of healthy and diseased plants. (A) Diagram of a healthy and diseased *P. cyrtonema* plant and the belowground compartments, including the rhizome soil, rhizosphere soil, zone soil, and furrow soil; H-rhizome, healthy rhizome; D-rhizome, diseased rhizome. (B) Principal-component analysis (PCA) of rhizome saponins and other metabolites of the healthy and diseased rhizomes. (C) Fresh weight of two samples; *t* test, *P* < 0.001. (D) Polysaccharide content of two samples; *t* test, *P* < 0.05. (E) Heatmap clustering of the relative content in rhizome saponins and other metabolites; *n* = 12.

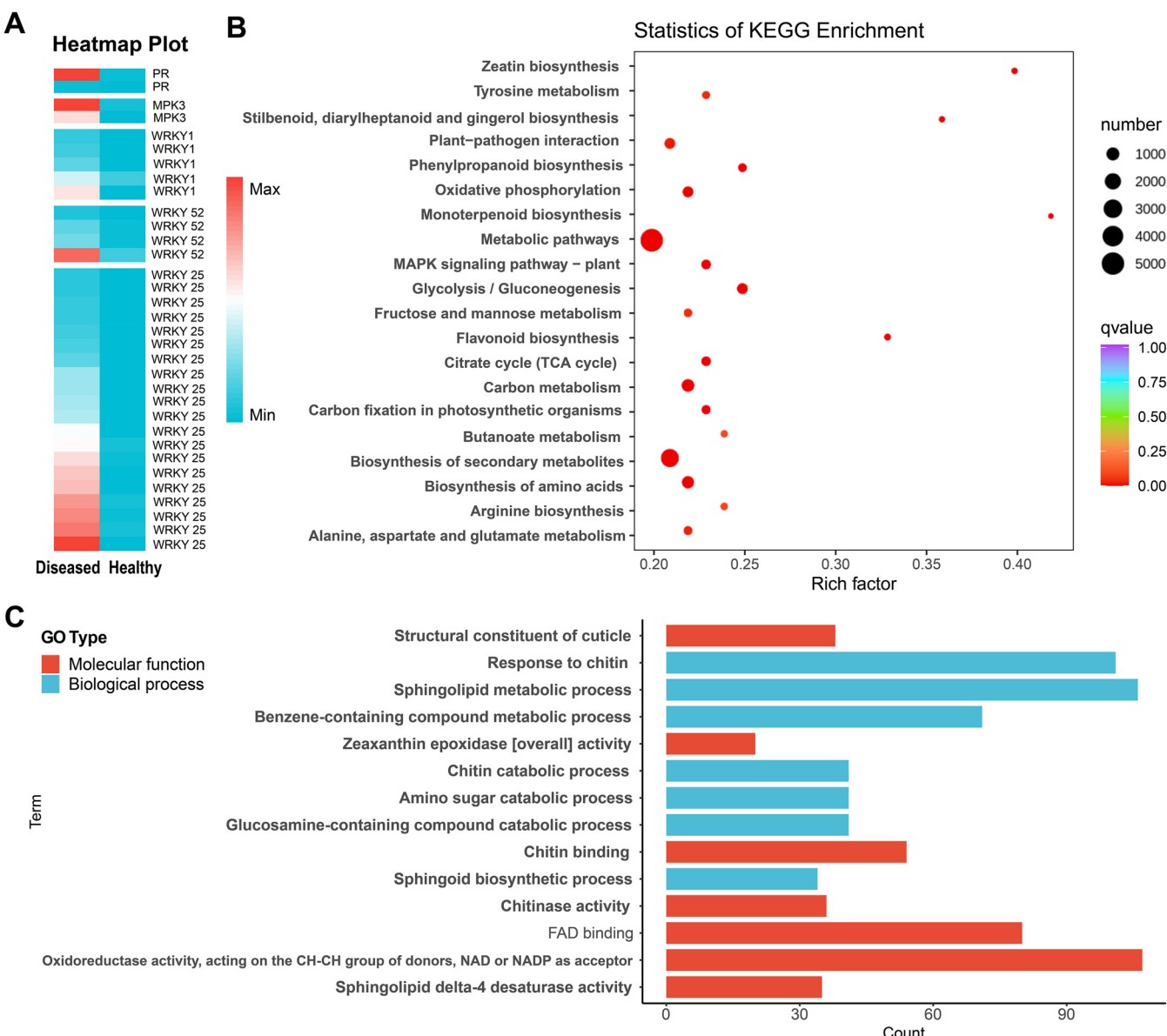

**FIG 2** Transcriptome analysis of healthy and diseased rhizomes. (A) Expression patterns of defense genes in healthy and diseased rhizomes; *n* = 12; genes with an FPKM of <10 expression were removed during heatmap generation. (B and C) Annotation and KEGG (top 20) (B) and GO (top 15) (C) pathway analysis of the identified DEGs.

Several disease resistance and defense synthesis-related GO terms were commonly found in the diseased rhizome (Fig. 2C), and these terms included "response to chitin (GO:0010200)," "flavonoid biosynthetic/metabolic process (GO:0009812/0009813)," "glucosamine-containing compound catabolic process (GO:1901072/1901071)," "sesquiterpenoid biosynthetic/metabolic process (GO:0016106/0006714)," "fructosyltransferase/6G-fructosyltransferase/beta-fructofuranosidase activity (GO:0050738/0033841/0004564)," and "cycloartenol synthase activity (GO:0016871, related to saponin synthesis)" (Fig. S2 and Table S2). Additional terms specifically associated with disease resistance, including "systemic acquired resistance, salicylic acid mediated signaling pathway (GO:0046224 and 0009696)," "induced systemic resistance/systemic acquired resistance (GO:0009682/0009627)," "response to hydrogen peroxide (GO:0042542)," "salicylic acid catabolic process (GO:0046244)," "jasmonic acid biosynthetic process (GO:0009867, 0009694 and 0009695)," "ethylene-activated signaling pathway (GO:0009873)," "abscisic acid metabolic process (GO:0009687)," and "MAP kinase activity (GO:0004708)," were also found in the diseased rhizomes.

An analysis of the enriched pathways in the KEGG analysis showed that the pathways that were most enriched under rhizome rot conditions were "biosynthesis of secondary metabolites (ko01110)," "glycolysis/gluconeogenesis (ko00010)," "flavonoid biosynthesis (ko00941)," and "MAPK signaling pathway-plant (ko04016)" (Fig. 2B). Consistent with the GO analysis, some important disease resistance and defense metabolite pathways, such as "plant-pathogen interaction (ko04626)," "plant hormone signal transduction (ko04075)," and "benzoxazinoid biosynthesis (ko00402)," were identified (Table S2). Importantly, the MAPK signaling pathway mitogen-activated protein kinase 1/2/3/6 (MPK1/2/3/6) (EC:2.7.11.24) and MAP kinase substrate 1 (MKS1) showed upregulated expression patterns in the diseased rhizomes, and these kinases are related to defense responses and stress adaptation. In addition, numerous genes were associated with plant-pathogen interactions, including genes associated with plant innate immunity and salicylic acid-responsive proteins that exhibited upregulated expression in the diseased rhizomes, such as pathogenesis-related protein 1 (PR1), auxin-responsive GH3 gene family (GH3), WRKY transcription factor 1/25/29/52 (WRKY 1/25/29/52), and other resistance genes (Fig. 2A). All 1,196 transcription factors (TFs; 52 gene families, 914 upregulated TFs) were identified among the DEGs; the *MYB*, *ERF*, *WRKY*, *C2H2*, and *NAC* families included the highest numbers of differentially expressed TFs in the diseased rhizomes compared with healthy controls (Table S2). These results revealed that the rhizome transcriptional and metabolic response to root rot in diseased plants increased the numbers and expression of disease resistance genes relative to those in healthy plants ($P < 0.01$).

**Root rot disrupts the *P. cyrtonema* rhizome microbial community and reduces diversity.** To identify the effect of root rot on the *P. cyrtonema* belowground microbiota and to identify potential pathogenic fungi and beneficial microbes, we sequenced bacterial and fungal amplicons from five compartments (rhizome, rhizosphere, zone, furrow, and no planted soil; $n = 51$) of six healthy and six diseased plants. A total of 5,298,508 high-quality reads (2,045,215 bacterial 16S rRNA and 3,253,293 fungal internal transcribed spacer [ITS]) were obtained from 48 healthy and diseased samples and 3 bulk soil samples. These reads were sorted into 9,540 bacterial operational taxonomic units (OTUs) and 4,685 fungal OTUs (Table S3). To visualize the similarity and dissimilarity in the microbial communities among the healthy and diseased samples, principal coordinate analysis (PCoA) and analysis of similarity (ANOSIM)/Adonis analysis were performed based on OTUs of ITS and 16S rRNA gene amplicon sequencing using the Bray-Curtis metric.

Notably, root rot reduced the rhizome alpha diversity of both the fungal and bacterial communities (Shannon index, Student's *t* test; Fig. 3A and B; for other alpha diversity indices, refer to Table S4). The PCoA revealed a clear clustering of the bacterial and fungal community compositions in the rhizomes from both the healthy and diseased samples according to the ANOSIM test (Fig. 3C and D; Table S5). The first principal coordinate of root rot conditions explained 65.35% (bacteria) and 51.94% (fungi) of the total variance (Fig. 3C and D). Healthy rhizome and bulk soil highly overlapped, indicating that the fungal and bacterial communities of these two soil compartments were generally similar.

The two-tailed Wilcoxon rank sum test was used to evaluate differences in the relative abundances of bacteria and fungi at the genus level between the healthy and diseased rhizomes (Table S6). A total of 143 differentially abundant bacterial genera were identified between the healthy and diseased rhizomes ($P < 0.05$). The community structure and abundance analysis also indicated an increased abundance of the fungal genera *Candida* (54.9%), *Clonostachys* (10.22%), and *Fusarium* (7.75%) and the bacterial genera *Rahnella* (24.34%), *Anaerosinus* (15.61%), *Clostridium* (5.66%), and *Gluconacetobacteria* (4.54%) in diseased rhizomes compared with healthy rhizomes (fungal genera 0.01%, 3.54%, and 3.01% and bacterial genera 0.01%, 0.01%, 0.01%, and 0.01%, respectively) (Fig. 3E and F). Furthermore, several bacterial taxa/genera, such as *Gaiellales* (4.22%), *Vicinamibacterales* (3.47%), and *Sphingomanas* (2.88%), and fungal genera *Mortierella* (15.12%) and *Saitozyma* (12.75%) were enriched in the healthy rhizome. The results described above revealed that

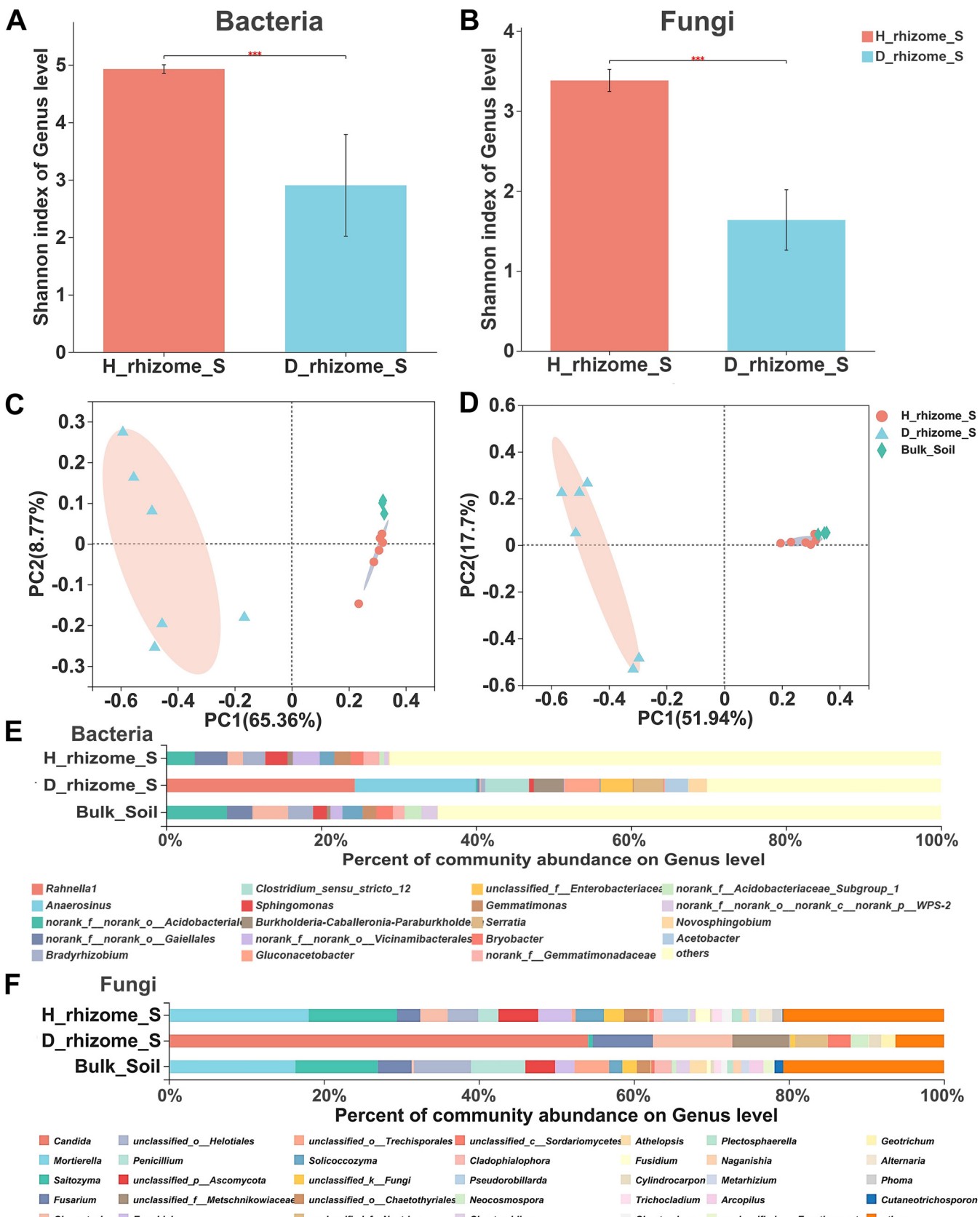

**FIG 3** Microbial community analysis of healthy and diseased rhizome soil. (A and B) Alpha-Shannon diversity indices of bacterial (A) and fungal (B) communities in the healthy and diseased *P. cyrtonema* rhizome soil. The healthy sample bacterial and fungal alpha indices were 4.9272 and 3.9818,

root rot changed the microbiota of *P. cyrtonema* present in the rhizome soil, with the strongest impact on the bacterial community.

**Root rot affects rhizosphere microbiota community and function.** In the micro-biome analysis, we found that rot had a negative effect on the rhizosphere microbiota, and this effect tended to be irrecoverable and persistent in plants. First, root rot affected the fungal and bacterial communities in the rhizomes to a larger extent than that in the rhizosphere (Fig. 4A to D). However, root rot had no significant effect on mi-crobial community structure and diversity in the zone and furrow soil (Fig. 4A to D; Fig. S3 and S4 and Table S5). All the samples were clustered by soil-root system compart-ment types (all different types of soil, ANOSIM; bacterial: $R = 0.39$, $P = 0.001$; fungal: $R = 0.41$, $P = 0.001$) and healthy and diseased conditions (all healthy versus all disease, ANOSIM; bacterial: $R = 0.15$, $P = 0.001$; fungal: $R = 0.23$, $P = 0.001$) (Table S5). For each compartment, fungal communities were more variable than bacterial communities in all the healthy and diseased samples. The Wilcoxon rank sum test was next used to explore the most important driver of microbial alpha diversity. The analysis revealed that the different compartments were the main factors influencing the alpha diversity of microbial communities based on Shannon and other diversity indices (Fig. 4A and B; Table S4). It is obvious that root rot reduced rhizome fungal diversity and changed the community structure (Wilcoxon rank sum test, $P < 0.001$; Fig. 4B; Fig. S3 and Table S4). Similarly, rhizosphere bacterial communities (ANOSIM; $R = 0.36$, $P = 0.003$) were less affected by root rot than fungi ($R = 0.62$, $P = 0.003$).

Taxa enrichment and depletion in the diseased plants were more pronounced than those in the healthy plants ($P < 0.001$; Fig. S3 and S4). The relative abundance of the fungal genus *Candida* was also significantly higher in the diseased rhizosphere than in the healthy rhizosphere ($P = 0.02$; Fig. 4F). Interestingly, several bacteria from the gen-era *Bradyrhizobium*, *Sphingomonas*, *Bryobacter*, and *Xanthobacteraceae* were signifi-cantly enriched in the healthy plants ($P < 0.05$; Fig. 4E). For the diseased plants, the bac-terial genera *Lactococcus*, *Enterobacter*, and *Rahnella* were the top three genera enriched in the diseased rhizosphere ($P < 0.01$). In addition, we used the BugBase phenotype predic-tion approach to explore the functional shift in the rhizosphere-associated microbiota that was potentially induced by rhizome rot. The relative proportions of the phenotype associ-ated with the potentially pathogenic and biofilm forms increased in the microbiota of the diseased rhizome and rhizosphere soil compared with the healthy controls ($P = 0.005$; Fig. S6 and Table S7). However, this phenotype was not significant in the root-zone soil and fur-row soil of healthy and diseased plants (Table S7). In other words, the function of the rhizo-sphere microbiota of diseased plants is altered by rhizome rot and becomes pathogenic, while the rhizosphere microbiota of healthy plants does not become pathogenic.

**Potential fungal pathogens and beneficial microbiota.** Based on the above microbiome sequencing results regarding the differences in the microbiota between healthy and diseased rhizomes and rhizosphere soil, pathogenic fungi were further iso-lated from the leave, stems, rhizomes, and soil of diseased plants. In addition, some potentially beneficial (biocontrol) bacteria were isolated from healthy rhizome endo-phytes and rhizosphere soil because they were different from those in diseased plants (Fig. 4E; Fig. S5A). The numbers of fungal species in each component were 59 in the leaf and stem, 88 in the diseased rhizome, and 66 in the diseased plant soil, and the most abun-dant fungal genera were *Fusarium* (23.47%), *Candida* (10.80%), and *Colletotrichum* (10.33%) (Fig. 5A; Table S8). The most abundant fungal species, *Fusarium oxysporum* (17.84%), *Candida* sp. (10.80%), *Colletotrichum spaethianum* (10.33%), and *Fusarium solani* (5.63%), were isolated from the diseased rhizome endophytes. Therefore, the pathogenicity of the four isolates described above was tested on 2-year-old *P. cyrtonema* plantlets. Yellow color,

**FIG 3** Legend (Continued)
respectively, and diseased sample bacterial and fungal alpha indices were 2.9047 and 1.8935, respectively, on average; $n = 12$, Student's *t* test, $P < 0.001$. (C and D) PCA of bacterial (C) and fungal (D) community beta diversity of the healthy and diseased rhizome soil; bacteria: $R = 0.9685$, $P = 0.003$; fungi: $R = 0.9241$, $P = 0.003$. (E and F) Bacterial and fungal genera community structure of the healthy and diseased rhizome soil. (E) Bacterial community (top 10 genera in relative abundance). (F) Fungal genera community.

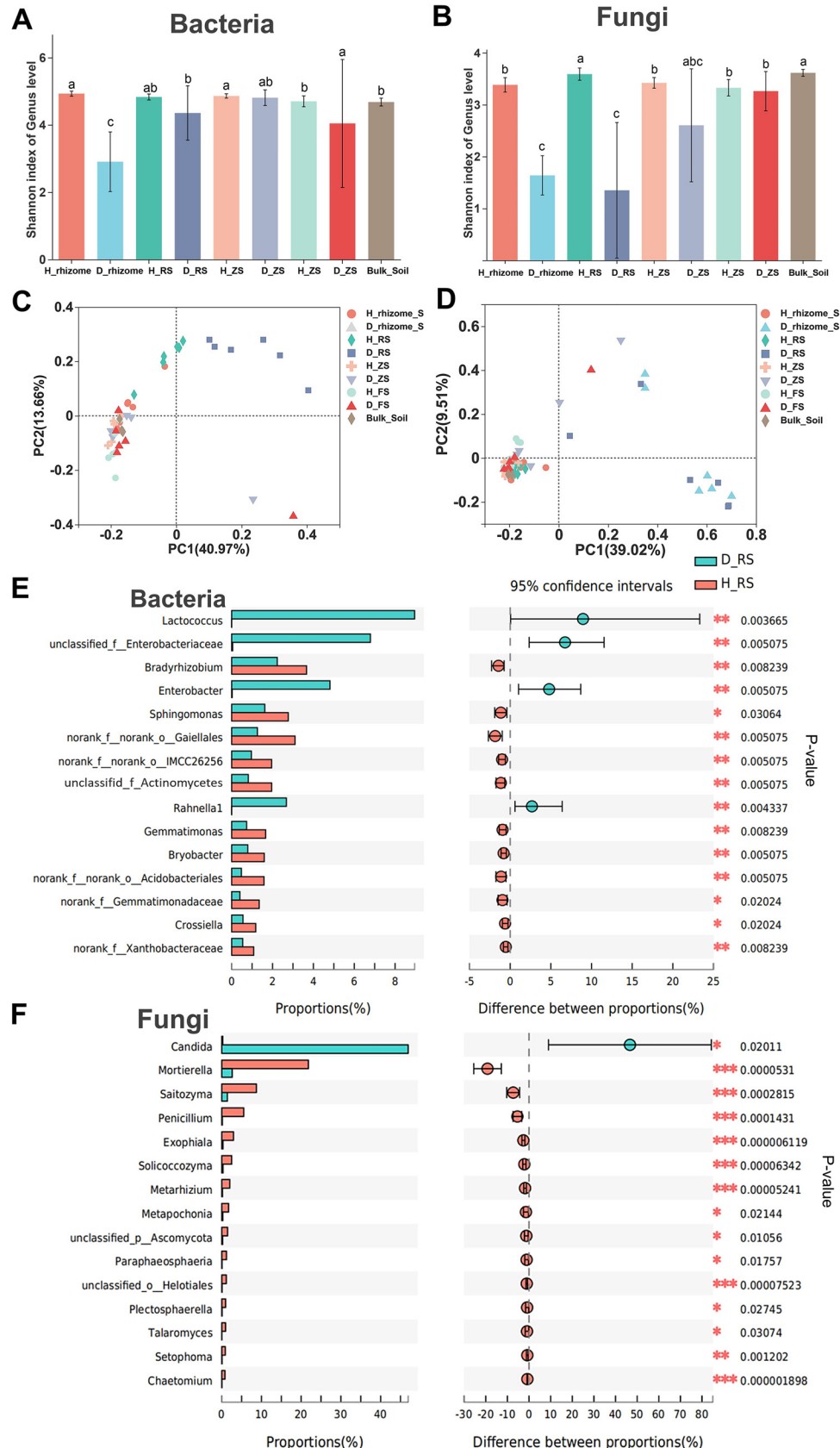

**FIG 4** Bacterial and fungal community and differential analysis of all healthy and diseased plant samples. (A and B) Alpha-Shannon diversity indices of bacterial (A) and fungal (B) communities of all soil samples; *n* = 27,

chlorosis, and withering of stems and leaves developed 2 weeks after the inoculation of plantlets with *Fusarium oxysporum* and *Colletotrichum spaethianum* isolates (Fig. 5B, panels b and c), and no obvious symptoms were observed after inoculation with *Candida* sp. or *Fusarium solani* isolates or on control plantlets (Fig. 5B, panels a, e, and f). The pathogen isolates that caused symptoms on the inoculated plants were reisolated and identified as *Fusarium oxysporum* and *Colletotrichum spaethianum* species.

After the identification of the pathogenic fungi that cause *P. cyrtonema* rhizome rot, we further isolated beneficial bacteria to investigate their function in the biocontrol of plant disease symptoms *in vitro* and *in vivo*. In total, 472 bacterial isolates were recovered from healthy plant rhizome endophytes and rhizosphere soil. Here, we focused on *Sphingomonas* ($P < 0.05$), *Actinomycetales* (Fig. 5C; $P < 0.01$), and *Xanthobacteraceae* ($P < 0.01$) because they were more abundant in healthy rhizomes and rhizospheres than diseased plant samples (Fig. 3E, Fig. 4E, and Fig. S5A). Among these, 22 isolates were identified as candidate biocontrol strains that belong to the *Streptomyces*, *Burkholderia*, *Pseudomonas*, and *Klebsiella* genera (Table S8 and Fig. S5), and their potential antifungal ability was shown in plate confrontation tests. Among the strains mentioned above, only the 16S rRNA of *Streptomyces violascens* XTBG45 (here referred to as HJB-XTBG45) had 100% nucleotide similarity with the 16S rRNA sequences of *Stenotrophomonas* sp. (Fig. 5D; Table S8). We further evaluated whether HJB-XTBG45 reduced *P. cyrtonema* symptoms, and we treated the *Fusarium oxysporum*-infected variety of *P. cyrtonema* rhizomes with HJB-XTBG45 in glasshouse experiments (Fig. 5B, panel d). Overall, inoculation with HJB-XTBG45 at the time of sowing conferred significant protection against *F. oxysporum* compared with the negative control (Fig. 5B, panel c). In addition, we are still verifying whether the HJB-XTBG45 strain alleviates the rhizome rot caused by other pathogens and green fluorescent protein (GFP)-tagged strains.

## DISCUSSION

Root rot (rhizome rot) is a severe pathogenic fungal disease that causes great damage and poses a severe threat to the development of the *P. cyrtonema* industry (6). However, the potential mechanisms underlying the defense system of *P. cyrtonema* plants in response to this disease as well as potential pathogenic and beneficial biocontrol microbes have not been clearly defined. In this study, three omics approaches, namely, metabolome, transcriptome, and microbiome analyses, were used to gain insights into potential molecular processes and beneficial microbiota related to the tolerance of *P. cyrtonema* plants to rhizome rot. Herein, our work provides evidence that rhizome rot not only changes rhizome transcription and functional metabolite contents but also impacts the microbial community diversity, assembly, and function of the rhizome and rhizosphere. Below, we discuss how these findings have advanced our understanding of the disease-induced changes in *P. cyrtonema* plant functional gene transcription, compound metabolism, and belowground microbiota assembly and functions.

**Diseased rhizome transcription and metabolism may be involved in the plant defense response.** The plant hormones salicylic acid (SA), jasmonate (JA), ethylene (ET), and auxin play central roles in plant resistance to pathogens via hormone signal transduction or the MAPK signaling pathway (37–40). It is generally thought that JA/ET and SA contribute to defense against some biotrophic/necrotrophic and biotrophic pathogens, respectively. In this study, the expression of various classes of SA, ET, WRKY, and MAPK, including *ERF1/2*, *PR1*, *WRKY*1/22/25/29/52, and so on, was increased in diseased rhizomes, and the expression of *JAR1* and *COI1* in the JA pathway was

**FIG 4** Legend (Continued)

Student's *t* test. (C and D) PCA of bacterial and fungal community beta diversity of all samples (healthy rhizome versus diseased rhizome, ANOSIM; bacterial: $R = 0.96$, $P = 0.003$; fungal: $R = 0.93$, $P = 0.003$) to a larger extent than that in the rhizosphere (healthy versus diseased rhizosphere, ANOSIM; bacterial: $R = 0.36$, $P = 0.003$; fungal: $R = 0.62$, $P = 0.003$; on average, respectively); healthy versus diseased plant, ANOSIM; zone and furrow bacterial: $R = 0.01$, $P = 0.385/R = -0.04$, $P = 0.674$; fungal: $R = 0.13$, ($P = 0.054/R = 0.1$, $P = 0.1564$; on average, respectively). (E and F) Significance test between healthy and diseased rhizosphere soil bacterial and fungal community groups; Wilcoxon rank sum test.

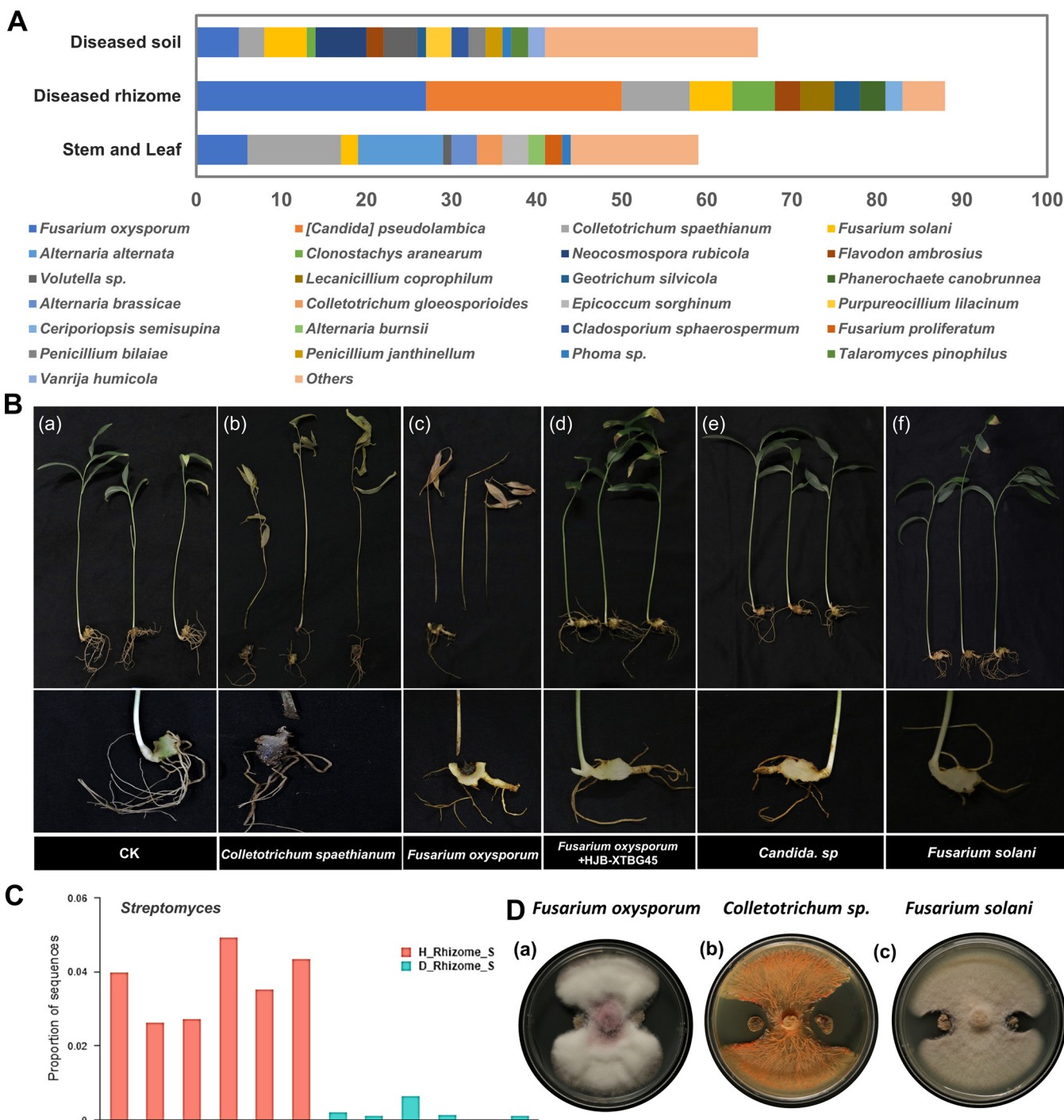

**FIG 5** Differentiation analysis of all healthy and diseased rhizome soil. (A) Isolated fungal communities in the diseased *P. cyrtonema* rhizome (top 25 strains). (B) Inoculation of different potential pathogens and the effect of HJB-XTBG45 on the *P. cyrtonema* rhizome. There were 5 biological replicates per treatment. (b and c) The plants of the inoculated strain showed disease characteristics similar to those of the field. (a and d to f) No disease characteristics were found in these plants. (a) CK; (b) inoculation of *C. spaethianum*; (c) inoculation of *F. oxysporum*; (d) inoculation of *F. oxysporum* and *Streptomyces violascens* XTBG45; (e) inoculation of *Candida* sp.; (f) inoculation of *F. solani*. (C) Relative abundance of *Streptomyces* between healthy and diseased rhizomes; Wilcoxon rank sum test and *t* test, *P* < 0.01. (D) Inhibitory effects of *Streptomyces violascens* XTBG45 (HJB-XTBG45) on *F. oxysporum*, *C. spaethianum*, and *F. solani*.

downregulated. These findings provide evidence that supports the idea that the genes mentioned above play key roles in plant defense mechanisms and may contribute to *P. cyrtonema* immunity. Furthermore, *P. cyrtonema* quality markers (Q-Marker), such as polysaccharides (and glycosidase) (23, 38, 40–42) and saponins (mainly steroidal and

triterpene saponins) (25, 43, 44), are natural antimicrobial compounds (or compounds required for the biosynthesis of antimicrobial defense compounds) that increase the resistance of plants to pathogens, and they are important primary and secondary metabolites. Interestingly, the expression of genes related to polysaccharide, flavonoid, and saponin metabolism, including glycolysis/gluconeogenesis, starch, sucrose, fructose, mannose, galactose, and isoflavonoid, was particularly induced in diseased rhizomes (Table S3 in the supplemental material), and this increased expression was accompanied by the accumulation of several triterpenes and monoterpenoids, such as 3$\beta$-hydroxy-28-norurs-17,19,21-trien, betulin, and 3′-*O*-ᴅ-glucosylgentiopicroside (Table S1). In particular, most polysaccharides and oligosaccharides trigger an initial oxidative burst at the local level and activate SA, JA, and/or ET signaling pathways, or these factors act to elicit responses to efficiently protect the plant against biotic stresses at the systemic level (45–47). Thus, we hypothesize that polysaccharides, saponins, and other metabolites might play important roles in the response of *P. cyrtonema* plants.

**Disease affected the *P. cyrtonema* rhizome and rhizosphere microbiota composition and functions.** In addition to inducing changes in plant transcription and metabolism, such as defense gene expression, hormone signaling, and functional metabolite contents, root rot also affects the microbial community in the *P. cyrtonema* rhizome and rhizosphere, and this effect may cause irreversible plant damage. Plant-pathogenic bacteria and fungi have long received attention because they can directly infect host plants and cause diseases in crops, but the role of the root-associated microbial community in diseased plants is of interest because the rhizosphere is the first line of pathogen defense and plays a key role in the prevention of pathogenic bacterial and fungal invasion (7, 36). In this study, we found that the bacterial and fungal communities in the diseased plants were significantly different and more variable than those in the rhizome and rhizosphere compartments of healthy plants.

The reduction in root-associated microbial diversity and the distinct changes in the microbial community were previously shown to be associated with disease occurrence (48). In general, a richer microbiota community composition has a stronger ability to control and reduce pathogens (49, 50). Highly diverse microbial communities tend to be more complex and possess greater functional redundancy and interkingdom associations (16, 51). Similar to our study, reduced microbial diversity was found in diseased bananas plants (52), and microbial diversity was found to be higher in healthy sugar beets (48), beans (50), and chili peppers (16). In addition, decreased diversity facilitates the invasion of potential pathogenic bacteria and fungi into soil or plant-associated microbial communities (53). Lower diversity in diseased plants was indicated by a significant decrease in bacterial fungal diversity indices (48). Furthermore, our results indicated that at the genus level, the most abundant bacteria and fungi enriched in the diseased plant rhizome and rhizosphere were *Rahnella* (24.34%) and *Candida* (54.9%), which was similar to a previous study reporting that these genera include biotrophic pathogens of diverse plant species (54, 55).

Furthermore, predictive function analyses indicated that microbiota involved in biofilm formation were enriched in the diseased plant rhizosphere compared with the healthy plant rhizosphere. Biofilms foster increased resistance to certain environmental stresses as well as tolerance to the antimicrobial mechanisms of diseased host plants. Bacterial biofilms confer the same fundamental advantages to pathogens as commensals; surface attachment is typically the first step in pathogenesis, and aggregation promotes virulence and protection against plant defense responses (56–58). Based on the functional prediction of the bacterial community to the categories of "forms biofilms" and "potential pathogens," we hypothesized that the rhizosphere microbiota reassembly caused by rhizome rot may cause irreversible plant disease. In conclusion, our study found that disease-resistant plants tend to maintain relatively higher diversity or evenness as well as relatively strong interactions in their rhizosphere microbiota than susceptible species and varieties.

**Is there a *P. cyrtonema* rhizosphere "cry for help" strategy in response to rhizome rot?** Previous studies have suggested the "cry for help" hypothesis, which states that the roots of diseased plants or isolated pathogenic fungi can selectively enrich plant-protective microbes to overcome or adapt to biotic and abiotic stress (59, 60). Some evidence for the "cry for help" hypothesis is beginning to accumulate; for example, in the defense activation of barley (61), beans (50), wheats (62), peppers (16), cucumbers (36), tomatoes (17), *Carex arenaria* (63), and *Arabidopsis* (18, 21), it has been demonstrated that beneficial bacteria contribute to host plant disease defense by priming the plant immune system, secreting antibiotic compounds, or competing for resources with the pathogen (16). Furthermore, host plants may attract beneficial microbiota by secreting (or modifying the synthesis of) root exudates (REs) or volatile organic compounds (VOCs) (31, 60, 64). For example, Gao et al. (16) showed that the beneficial bacterium *Stenotrophomonas rhizophila* could enhance pepper plant defenses in the stems and fruits when the soil pathogen *Fusarium pseudograminearum* was present. However, in this study, we found that bacteria, such as *Rahnella*, *Enterobacter*, and *Lactococcus*, that were enriched in the diseased plant rhizosphere were also enriched in the diseased rhizome compared with the healthy rhizosphere, and these bacteria may further cause disease in the host plant (Fig. 5; Fig. S3). These bacteria did not suppress or control *Fusarium* and other pathogenic fungi in the plate confrontation test. Moreover, our results indicated that healthy plants may selectively regulate the community abundance of some potential beneficial bacterial taxa, such as *Bradyrhizobium*, *Sphingomonas*, and Xanthobacteraceae, and fungal genera, such as *Candida* and *Clonostachys*. The empirical evidence supporting a "cry for help" strategy in our study is limited. Plant species differ in physiological and immune responses to different pathogen invasions, meaning that the functional metabolic basis and molecular mechanisms underlying the "cry for help" strategy may be dependent on different plant species. However, experimental evidence for the "cry for help" strategy is still rare, and the underlying plant and microbial mechanisms are still not clear (62).

**Potential pathogens and beneficial bacteria.** In recent years, researchers have described the important role of the microbiota in plant disease resistance. In this study, potential pathogenic fungi (*Fusarium* and *Colletotrichum*) and beneficial bacteria (*Streptomyces*) were identified by microbial amplicon sequencing and isolation as well as by inoculation experiments of healthy and diseased plants and samples. *F. oxysporum*, which causes *Fusarium* vasculature (wilt diseases) in more than 100 host plants (65), challenges the production of a large amount of economic crops, such as tomatoes, cotton, and bananas (66). Gordon et al. reviewed the knowledge on *Fusarium* root diseases, such as *Fusarium* yellows, *Fusarium* blight, and *Fusarium* wilt (67). Pathogenic *F. oxysporum* shows vascular wilting and results in root and plant rot. *F. oxysporum* causes host plant vascular wilting to penetrate the root and xylem vessels, resulting in further wilting of the crops (68). We are studying this phenomenon further using a GFP-labeled strain to research the mode of invasion of *F. oxysporum* into the rhizome and its effect on the *P. cyrtonema* plant. Similarly, *Colletotrichum* is also a large genus that causes destructive diseases in numerous crops, such as species of *Brassica* and *Raphanus* (69). Presently, leaf spots caused by *C. spaethianum* in *Polygonatum odoratum* and *Polygonatum cyrtonema* were reported from the Jilin and Anhui Provinces in China (70, 71). To date, our study is the first to report that *C. spaethianum* infects *P. cyrtonema* and causes rhizome rot in the Guizhou Province, China. On the other hand, our study showed that the diseased plants were more enriched in other fungi, such as *Candida* and *Clonostachys*, and *Candida* is recognized as a potential genus that suppresses pathogenic fungi and is used in controlling plant diseases (72, 73). Therefore, more work needs to be conducted with further inoculation of *P. cyrtonema* crops.

Actinomycetes (*Streptomyces* species) are well known to have great potential to be used as excellent agents for controlling various pathogenic fungi and bacteria because of their wide spectrum of antibiotics. *Streptomyces* spp. belong to the rhizosphere and soil microbial communities and are more abundant colonizers of plant roots and other tissues (74–76). Herein, *Streptomyces violascens* XTBG45 (HJB-XTBG45) isolated from the endophyte of the *P. cyrtonema* rhizome may synthesize a range of enzymes associated with defense against fungal pathogens when inoculated on diseased *P. cyrtonema*. In addition, a recent

study showed that *Streptomyces* spp., from the soil can accumulate on the leaf surface to induce host immune responses and against fungal pathogens (77). Overall, our findings suggested that the use of HJB-XTBG45 can potentially be expanded to manage root rot of *P. cyrtonema* rhizomes.

## MATERIALS AND METHODS

**Field trial and treatments.** Healthy and diseased *P. cyrtonema* samples were collected from plantations in Liu Panshui (LPS; 26°33′20″N, 104°45′50″E; altitude 2,022 m), Guizhou Province (Southwest China), in July 2021. In November 2020, 2-year-old *P. cyrtonema* rhizomes were surface sterilized (60 min in 1‰ carbendazim and rinsed with tap water three times). Sterilized *P. cyrtonema* rhizomes were grown in a random arrangement in two nearby fields (Fig. S1 in the supplemental material). The two fields were approximately 80 m long and 12 m wide, there were 10 ridges lateral to the longest side that were approximately 30 cm high, and nearly all ridges were 40 cm apart. The rhizomes were planted in adjacent rows (on the ridges) with a separation of 50 cm. The sampled field had a maize, bean, and other crop planting history of many years.

**P. cyrtonema material and soil sample collection.** Samples were collected on 7 July 2021 (8 months after planting, and diseased plants turned yellow and had wilting symptoms at about 30 days; Fig. S1). Six replicates of healthy and diseased plants were randomly collected from 10 ridges in two fields. Healthy (H_sample) and diseased (D_sample) plant rhizomes were collected from 6 samples for transcriptome, polysaccharide, saponin, and other metabolite analyses. Briefly, diseased and healthy rhizomes were immediately frozen in liquid nitrogen after removing aboveground tissues as well as root and rhizome surface soil. With the microbiome study, four soil-rhizome and root compartments, topsoil of the rhizomes (H/D_Rhizome_soil), rhizosphere soil (H/D_RS), root-zone soil (H/D_ZS), and furrow soil (H/D_FS) of healthy and diseased plants were analyzed. Unplanted soil (bulk soil) was used for soil physical and chemical properties and microbial sequencing analysis. All rhizomes and soil samples of each plant were quickly frozen in liquid nitrogen and dry ice transferred to the laboratory refrigerator ($-80°C$) for rhizome metabolomic and transcriptome sequencing analysis (the diseased rhizome used for pathogen isolation was placed in a sterile Ziploc bag to be taken back to the laboratory for treatment).

**P. cyrtonema rhizome RNA isolation and transcriptome analysis.** Total RNA was extracted from diseased and healthy rhizomes using a Pure RNA isolation kit (Tiangen, China) following the protocol. The prepared cDNA libraries were sequenced on the Illumina sequencing 2500 platform by Metware Biotechnology Co., Ltd. (Wuhan, China). Transcriptome sequencing analysis of 12 samples was completed in this study; 80.73 Gb of clean data were obtained using the Illumina platform, and the percentage of Q30 bases was above 90%.

**P. cyrtonema rhizome polysaccharide and saponin metabolite profiling.** The relative quantities of polysaccharides, saponins, and other metabolites in healthy and diseased rhizome samples were analyzed with an ultrahigh-performance liquid chromatography-electrospray ionization-tandem mass spectrometry (UPLC-ESI-MS/MS; used for saponins and other metabolites) system, and polysaccharide contents were detected based on an Agilent 7890B gas chromatograph coupled to a 7000D mass spectrometer (GC-MS) platform by MetWare (Wuhan, China) (see supplementary methods 1 and 2 for details regarding the protocol). A total of 44 steroidal and triterpenoid saponins as well as other metabolites and 13 sugar metabolites were detected by the UPLC-ESI-MS/MS platform and GC-MS system based on the MetWare self-built database.

**Soil DNA extraction, 16S rRNA, and ITS gene amplification.** Healthy and diseased plant rhizome soil, RS, ZS, FS, and bulk soil DNA were extracted using the E.N.Z.A. soil DNA kit (Omega, USA) according to the manufacturer's protocol, and sample DNA quantity was surveyed using a NanoDrop 2000 spectrophotometer (Thermo Scientific, USA). The fungal ITS1 region (ITS1F/ITS2R) (78) and the bacterial 16S rRNA gene (V3-V4, 338F/806R) (79) were amplified (primer sequences and PCR amplification conditions were the same as those described in the reference). Amplicon libraries were sequenced on the Illumina MiSeq PE300 platform by the Majorbio Bio-Pharm Technology Co. Ltd. (Shanghai, China).

**Isolation, identification, and inoculation of bacteria and pathogenic fungi.** Potential pathogenic fungi isolated from the diseased rhizome were placed onto potato dextrose agar (PDA; Hopebio, Qingdao) and incubated for 7 days (in the dark at 25°C). Five grams of healthy rhizome pieces was weighed into a 50-mL centrifuge tube, and 100 $\mu$L of $10^{-3}$ and $10^{-4}$ diluents was soaked up with diluted coating plates onto tryptic soy agar (TSA; Hopebio, Qingdao) and international streptomyces project-2 medium (ISP2, Hopebio, Qingdao) to isolate potential biocontrol bacteria. Pure culture isolates were obtained by the streak plate or single-spore technique. ITS and 16S rRNA genes of the isolated strains were amplified with fungal primers ITS1F (5′-CTTGGTCATTTAGAGGAAGTAA-3′) and ITS4R (5′-TCCTCCGCTTATTGATATGC-3′) and bacterial universal 27F (5′-GAGAGTTTGATCCTGGCTCAG-3′) and 1492R (5′-ACGGATACCTTGTTACGACT-3′). ITS and 16S sequences were aligned with the NCBI ITS (fungi) and 16S rRNA sequences (bacteria and archaea) databases by nucleotide BLAST (https://blast.ncbi.nlm.nih.gov/) to determine the approximate phylogenetic affiliation. MEGA8 software was used to visualize the phylogenetic relationships using the neighbor-joining method by constructing a phylogenetic tree. To verify pathogenicity, 1-year-old *P. cyrtonema* seedlings were inoculated with fungal spore suspension by injurious rhizome inoculation for verification. There were 6 replicates in the control group and potential fungi treatment group. Typical wilting symptoms appeared 15 days after inoculation with fungi and were similar to diseased plants observed in the field, while the control group (CK) remained healthy. Moreover, the pathogenic fungi were isolated from the diseased rhizome, and ITS sequencing of these fungi was performed again (passing Koch's postulates).

**Bioinformatic analysis.** Trinity (v2.11.0) and Corset were used to assemble the transcriptome and regroup relevant transcripts into "gene" clusters (https://github.com/trinityrnaseq/trinityrnaseq). A basic analysis of transcriptome data, including data preprocessing, genomic mapping, gene expression analysis, analysis of differentially expressed genes (DEGs), Gene Ontology (GO), clusters of orthologous groups of proteins (COG), and Kyoto Encyclopedia of Genes and Genomes (KEGG) enrichment, followed reference 80. Expression levels were estimated based on the number of fragments per kilobase of transcript per million reads mapped (FPKM). Refer to supplementary method 3 "Bioinformatic and statistical analysis" for details.

The 16S rRNA and ITS gene sequencing reads were demultiplexed by Fastp (version 0.20.0) (81) and merged with FLASH (version 1.2.7) (82), and operational taxonomic units (OTUs) with 97% similarity were clustered using UPARSE (version 7.1) (83). Taxonomic assignment was performed using the bacterial SILVA reference database (v12_8,) and fungal UNITE database (v7.0). Functional prediction (Tax4Fun, PICRUst1/2, BugBase phenotype prediction) was analyzed on the free online platform of the Majorbio cloud platform (www .majorbio.com).

**Statistical analysis.** The corrected *P* values were used as the threshold for significantly differentially expressed genes. The alpha diversity, including the Shannon, Chao1, and Simpson indices, was determined using Mothur v. 1.34.4. The base R package "stats" (v. 3.4.1) was used to perform the two-tailed Wilcoxon rank sum test (wilcox.test function). Analysis of Adonis and ANOSIM (Bray-Curtis metric) was calculated with the "vegan" R package. The PCoA was performed to examine the similarities and dissimilarities within the healthy and diseased plant groups (based on Bray-Curtis dissimilarities retrieved from normalized OTUs). Some data and analyses, such as Spearman's correlations and others, were analyzed on the free online platform of the Majorbio cloud platform (www.majorbio.com).

**Data availability.** The raw sequence and other related data reported in this paper have been deposited in the National Genomics Data Center (NGDC) under accession code BioProject PRJCA003178 (Genome Sequence Archive: CRA005765; https://ngdc.cncb.ac.cn/).

## SUPPLEMENTAL MATERIAL

Supplemental material is available online only.
**SUPPLEMENTAL FILE 1**, XLSX file, 0.02 MB.
**SUPPLEMENTAL FILE 2**, XLS file, 15.2 MB.
**SUPPLEMENTAL FILE 3**, XLS file, 3.3 MB.
**SUPPLEMENTAL FILE 4**, XLSX file, 0.02 MB.
**SUPPLEMENTAL FILE 5**, XLSX file, 0.01 MB.
**SUPPLEMENTAL FILE 6**, XLSX file, 0.4 MB.
**SUPPLEMENTAL FILE 7**, XLS file, 0.01 MB.
**SUPPLEMENTAL FILE 8**, XLSX file, 0.04 MB.
**SUPPLEMENTAL FILE 9**, PDF file, 0.9 MB.

## ACKNOWLEDGMENTS

This study was supported by the Science and Technology Support Project of Chinese Academy of Sciences: Demonstration Project of Poverty Alleviation and Rural Revitalization Strategy in Shuicheng District, Guizhou Province (grant number KFJ-FP-202001), Research and Demonstration project of Agricultural and Rural High-Quality Transformation and Development in Shuicheng District, Guizhou Province (grant number KFJ-FP-202103), and Basic Research Funds of Hainan Academy of Agricultural Sciences (Open Project of Key Laboratory for Crop Breeding of Hainan Province, grant number 2021-05). We thank Cui Zhang (Yunnan Agricultural University) and Wenting Wang (Xishuangbanna Tropical Botanical Garden, Chinese Academy of Sciences) for their assistance in pathogen isolation experiments and data analysis and Lingfei Hu and Yachun Xu (Zhejiang University) for their help in writing the manuscript and drawing *P. cyrtonema* rhizome pattern diagrams. We thank the Wuhan Metware Biotechnology Co., Ltd. (Wuhan, China), for support during metabolite data analysis.

G.L., P.X., and Y.X. conceived the research and revised the manuscript. Z.P. completed all the sampling and measurement work. X.M. and J.X. participated in the bacterial and fungal separation experiment. X.M. and X.W. coordinated the literature research, and Z.P. drafted the initial version of the manuscript. All authors contributed to reviewing and finalizing the manuscript.

We declare no conflict of interests.

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
