## [Reviewer comments · Microbiology Spectrum]

Microbiology Spectrum

Multimomics reveals the effect of root rot on polygonati rhizome and identifies pathogens and biocontrol strain

Pang Zhiqiang, Mao Xinyu, Xia Yong, Xiao Jinxian, Wang Xiaoning, Xu Peng, and Liu Guizhou

Corresponding Author(s): Liu Guizhou, Xishuangbanna Tropical Botanical Garden, Chinese Academy of Science

Review Timeline:

Submission Date:	November 29, 2021
Editorial Decision:	December 16, 2021
Revision Received:	January 11, 2022
Editorial Decision:	January 23, 2022
Revision Received:	February 1, 2022
Accepted:	February 10, 2022

Editor: Yonglin Wang

Reviewer(s): The reviewers have opted to remain anonymous.

Transaction Report:

DOI: <https://doi.org/10.1128/spectrum.02385-21>

December 16, 2021

Prof. Liu Guizhou
Xishuangbanna Tropical Botanical Garden, Chinese Academy of Science
Mengla
China

Thank you for submitting your manuscript to Microbiology Spectrum. I received comments from two expert in this field and they have suggested some major modification to your ms before publication. Therefore, I invite you to respond the reviewers' and my comments and revise your ms. When submitting the revised version of your paper, please provide (1) point-by-point responses to the issues raised by the reviewers as file type "Response to Reviewers," not in your cover letter, and (2) a PDF file that indicates the changes from the original submission (by highlighting or underlining the changes) as file type "Marked Up Manuscript - For Review Only". Please use this link to submit your revised manuscript - we strongly recommend that you submit your paper within the next 60 days or reach out to me. Detailed instructions on submitting your revised paper are below.

(The authors represented interesting omics data to understand the association of root rot of Polygonatum with microbiota. My major concerns are as follows

1. Please clarify pictures of the root rot following inoculation of potential pathogens.
2. Would you conclude what biotic factors resulted in root rot from your isolation and amplicon analysis.
3. English writing needs to be polished.)

Link Not Available

Sincerely,

Yonglin Wang

Journals Department
Reviewer comments:

Reviewer #1 (Comments for the Author):

This manuscript by Pang et al. presented multi-omics study in Chinese traditional medical is very interesting. Overall, this manuscript deserves to be considered for publication after some revision, which hopefully can be helpful for the further improvements.

Major comments:

1. The figures quality is low, please carefully adjust the details and keep the font size and style keep the same. Such as, Fig. 1E "Z-Score" should be "Z-score", Fig. 2C legend to larger, top 50 in Fig.2 is too many (less than 30 items), Fig. 3E/F too many legends, Fig 5A too many legends, Letter stat method replace the line and star in fig. 4 A/B, Fig. 4F "Pvalue" should be "P-value", and so on.

2. The legend should describe the statistical method and replicate number (n).
3. Why in Fig. 3E the total relative abundance is not 1 or 100, and the x label is percent should the total be 100, not 1.

Minor comments:

1. Line 114 - 115, the 444.27 g is for healthy plants, 257.22 g should belong to root rot. Please check all the data in full text.
2. L144, L22, "significantly" should add the statistical method and P-value. Check the full text.
3. L202-203, Add "The first principal coordinate" before "Root rot conditions"

Reviewer #2 (Comments for the Author):

With multi-omics of plant-targeted metabolomics and transcriptomics, microbiome and culture-based methods, the manuscript of "Multiomics reveals the effect of root rot on polygonatum rhizome and soil microbiota assembly" investigated the metabolic and systemic resistance pathways as well as microbial composition and diversity between healthy and diseased Polygonatum plants. In addition, authors isolated potential bio-control agent *Streptomyces violascens* from healthy plants, which would inhibit pathogens of *Fusarium oxysporum* and *Colletotrichum spaethianum*. The manuscript aimed to investigate the interactions between the Polygonatum cyrtoneura plant, root-associated microbiota and pathogens (in Line 24). However, after examining the whole manuscript, I still have the following major issues to propose:

1) How to link the plant-targeted metabolomics and transcriptomics results with the occurrence of root rot disease of Polygonatum plants as well as the microbial communities?

Whether the metabolomics and transcriptomics results of healthy plants can inhibit the pathogen invasion, or the pathogen invasion on plants induced the difference of plant- metabolomics and transcriptomics? Which still need to be declared.

2) In line 38, "Taken together, our results indicate that *P. cyrtoneura* can modulate the plant immune system and metabolic processes and enrich beneficial microbes (rhizome and rhizosphere resistance) to defend against pathogens". How to evaluate the plant immune system in current study, based on the metabolomics and transcriptomics results?

3) Which ONE is the key pathogen for root rot disease of Polygonatum? *F. oxysporum*, *C. spaethianum* and *F. solani*? And whether they have joint effects on the root rot disease?

4) As shown in Figure 1, the relative contents of rhizome saponins and other metabolites were lower in healthy plants when compared with diseased ones, how to explain it? Do those metabolites can induce plant pathogens?

5) Microbiota assembly? Which kind of microbial assembly strategy in diseased and healthy plants, stochastic or deterministic, R or K strategy?

Other issues:

1) English should be improved by native speaker within the field. Please check the spelling mistakes and grammar throughout the whole manuscript.

2) Provide detailed information in Materials and Methods, for example Filed experiments description, and especially for the Bioinformatic analysis section.

3) Please carefully revise the discussion section.

Staff Comments:

Preparing Revision Guidelines

Please return the manuscript within 60 days; if you cannot complete the modification within this time period, please contact me. If you do not wish to modify the manuscript and prefer to submit it to another journal, please notify me of your decision immediately so that the manuscript may be formally withdrawn from consideration by Microbiology Spectrum.

This manuscript by Pang et al. presented multi-omics study in Chinese traditional medical is very interesting. Overall, this manuscript deserves to be considered for publication after some revision, which hopefully can be helpful for the further improvements.

Major comments:

1. The figures quality is low, please carefully adjust the details and keep the font size and style keep the same. Such as, Fig. 1E “Z-Score” should be “Z-score”, Fig. 2C legend to larger, top 50 in Fig.2 is too many (less than 30 items), Fig. 3E/F too many legends, Fig 5A too many legends, Letter stat method replace the line and star in fig. 4 A/B, Fig. 4F “Pvalue” should be “*P*-value”, and so on.
2. The legend should describe the statistic method and replicate number (n).
3. Why in Fig. 3E the total relative abundance is not 1 or 100, and the x label is percent should the total is 100, not 1.

Minor comments:

1. Line 114 – 115, the 444.27 g is the healthy plants, 257.22 g should belong to root rot. Please check all the data in full text.
2. L144 , L22, “significantly” should add the statistic method and P-value. Check the full text.
3. L202-203, Add “The first principal coordinate” before “Root rot conditions”

Dear Prof. Yonglin and Reviewers,

We sincerely thank you for your expert evaluation and specified comments on our manuscript, which will definitely help us to improve the quality of the manuscript. We have studied the comments carefully and revised the manuscript following your suggestions, which we hope will be met with approval. The main corrections in the paper and the responses to the reviewers' comments are as follows:

Responses to the editor's comments:

Prof. Yonglin Wang: The authors represented interesting omics data to understand the association of root rot of *Polygonatum* with microbiota. My major concerns are as follows:

1. Please clarify pictures of the root rot following inoculation of potential pathogens.

Response: In the part "Potential fungal pathogens and beneficial microbiota", we proved the potential pathogenic fungi and found potential biocontrol bacteria through microbiome sequencing, culture and isolation experiments. In the above process, we inoculated the top 4 fungi isolated from the diseased plants, such as *Fusarium oxysporum* (17.84%), *Candida sp.* (10.80%), *Colletotrichum spaethianum* (10.33%) and *Fusarium solani* (5.63%). However, diseased characteristics were observed after the inoculation of plantlets with *Fusarium oxysporum* and *Colletotrichum spaethianum* isolates (Fig. 5B-b and c), and no obvious symptoms were observed after inoculation with *Candida sp.* or *Fusarium solani* isolates or on control plantlets and rhizome segments (Fig. 5B-a, e and f).

In Fig. 5B, we further elucidated the specific phenotypic characteristics. "B Inoculation of different potential pathogens and the effect of HJB-XTBG45 on the *P. cyrtoneura* rhizome. (b and c) The plants of the inoculated strain showed disease characteristics similar to those of the field; (a, d-f) No disease characteristics were found in these plants. (a) CK, (b) inoculation of *C. spaethianum*, (c) inoculation of *F. oxysporum*, (d) inoculation of *F. oxysporum* and *Streptomyces violascens* XTBG45, (e) inoculation of *Candida sp.*, (f) inoculation of *F. solani*;"

2. Would you conclude what biotic factors resulted in root rot from your isolation and amplicon analysis.

Response: When setting up the experiment, we wanted to search for potential pathogens through the difference in microbiome sequencing between diseased and healthy plants and combine them with the culture results, focusing on the results of actual inoculation. By microbiome sequencing, we found that in diseased plants, the bacterial and fungal communities are disordered compared with healthy plants, and there are some advantageous microbes, such as yeast (*Candida sp.*) Then we inoculated the top four fungi in abundance isolated from the diseased plants, such as *Fusarium oxysporum* (17.84%), yeast *Candida sp.* (10.80%), *Colletotrichum spaethianum* (10.33%) and *Fusarium solani* (5.63%). Through the inoculation experiment, we found that *Fusarium oxysporum* and *Colletotrichum spaethianum* were the main pathogenic fungi, but yeast, which was the main fungus by sequencing, was not pathogenic (But the main advantage of the diseased plants fungus by microbiome sequencing (Fig. 5B-e)). We speculated that the invasion of pathogens may result in a decline in the plant's defense ability against microbes and the death of the plant. Yeast and other nutrition-utilizing microbes proliferate in large numbers, but most of these microbes do not cause disease (There were 5 biological replicates per treatment. Fig. 5B-e). Unfortunately, we did not sample the microbes in the stages of plant growth and diseased. We will continue to pay attention to the dynamic changes of microbiota in the process of pathogen invasion in the future studies. However, *Fusarium oxysporum* and *Colletotrichum spaethianum* are the main pathogenic fungi that cause *Polygonatum cyrtoneura* Hua root rot.

3. English writing needs to be polished.

Response: Thank you very much for your comments. We further improved the manuscript writing by working with a language editing service company.

Responses to the reviewers' comments:

Reviewer #1: This manuscript by Pang et al. presented multi-omics study in Chinese traditional medical is very interesting. Overall, this manuscript deserves to be considered for publication after some revision, which hopefully can be helpful for the further improvements.

Response: Thank you very much for all of your comments and suggestions.

Major comments:

1. The figures quality is low, please carefully adjust the details and keep the font size and style keep the same. Such as, Fig. 1E "Z-Score" should be "Z-score", Fig. 2C legend to larger, top 50 in Fig.2 is too many (less than 30 items), Fig. 3E/F too many legends, Fig 5A too many legends, Letter stat method replace the line and star in fig. 4 A/B, Fig. 4F "Pvalue" should be "P-value", and so on.

Response: Thank you very much for your comments. We have completely adjusted the format of the figures and reduced the legends of the figures.

2. The legend should describe the statistic method and replicate number (n).

Response: We sincerely thank the reviewer for this comment. We have added statistic method and repetition replicate number.

3. Why in Fig. 3E the total relative abundance is not 1 or 100, and the x label is percent should the total is 100, not 1.

Response: We sincerely thank the reviewer for this comment. We have revised this part (1=100%, thus, 0-1 is 0%-100%).

Minor comments:

1. Line 114 - 115, the 444.27 g is the healthy plants, 257.22 g should belong to root rot. Please check all the data in full text.

2. L144 , L22, "significantly" should add the statistic method and P-value. Check the full text.

3. L202-203, Add "The first principal coordinate" before "Root rot conditions"

Response: We sincerely thank the reviewer for this comment. We have revised it in the R1 manuscript.

Reviewer #2: Reviewer #2 (Comments for the Author):

With multi-omics of plant-targeted metabolomics and transcriptomics, microbiome and culture-based methods, the manuscript of "Multiomics reveals the effect of root rot on polygonati rhizome and soil microbiota assembly" investigated the metabolic and systemic resistance pathways as well as microbial composition and diversity between healthy and diseased Polygonatum plants. In addition, authors isolated potential bio-control agent Streptomyces violascens from healthy plants, which would inhibit pathogens of Fusarium oxysporum and Colletotrichum spaethianum. The manuscript aimed to investigate the interactions between the Polygonatum cyrtoneura plant, root-associated microbiota and pathogens (in Line 24). However, after examining the whole manuscript, I still have the following major issues to propose:

1) How to link the plant-targeted metabolomics and transcriptomics results with the occurrence of rood rot disease of Polygonatum plants as well as the microbial communities? Whether the metabolomics and transcriptomics results of healthy plants can inhibit the pathogen invasion, or the pathogen invasion on plants induced the difference of plant-metabolomics and transcriptomics? Which still need to be declared.

Response: Thank you for asking this interesting question. First, we focused on two metabolites: saponins and polysaccharides. These two metabolites are not only medicinal functional components of Polygonatum but may also be involved in the plant defense system as defense compounds (in lines 76-81: Previous studies have shown that plant metabolites and microbiota contribute to host plant defense against pathogens (Pang et al., 2021; Rodriguez et al., 2019; Song et al., 2021; Yuan et al., 2018). For example, previous studies have summarized the important role of polysaccharide (Barnes and Anderson, 2018; Gigli-Bisceglia et al., 2020) and steroidal and triterpenoid saponin toxins (Orme et al., 2019; Zaynab et al., 2021) in disease resistance.). In addition, the transcriptome can be combined with metabolomics analysis to study the synthesis of the above two functional metabolites and the defense progress of plants against pathogens. Furthermore, microbiome sequencing and culture-based methods can help us further identify potential pathogens and functional microbes.

In general, omics approaches can systematically elucidate the interactions of plants and other microbes with pathogens. Therefore, in this study, we surveyed Polygonatum plants naturally infected by fungal or bacterial pathogens and the defense signaling pathways in the field study.

In addition, another question is whether pathogen invasion is caused by changes in plant metabolites and transcription or changes in plant transcription and metabolism after pathogen invasion. First, it can be confirmed that the selected plants, soil and other environmental factors in the planting area are relatively consistent, and the physiological and metabolic activities of plants are relatively consistent, which can be reflected in the transcriptome and metabolic analysis results of healthy samples. However, due to the invasion of pathogens, the metabolic process of plants has changed, mainly reflected in the change of metabolite content and the difference of the transcriptome in MAPK, plant-pathogen interaction, plant hormones and other pathways. In discussion part: "Diseased rhizome transcription and metabolism may be involved in the plant defense response", we also elaborate and discuss this part in detail. In general, the invasion of pathogens changes the metabolic process of plants.

2) In line 38, "Taken together, our results indicate that P. cyrtonea can modulate the plant immune system and metabolic processes and enrich beneficial microbes (rhizome and rhizosphere resistance) to defend against pathogens". How to evaluate the plant immune system in current study, based on the metabolomics and transcriptomics results?

Response: Yes, we found in the transcriptome sequencing results that the differentially expressed genes (DEGs) between the diseased and healthy samples were mainly concentrated in mitogen-activated protein kinase (MAPK) signaling, plant-pathogen interaction, and plant hormone signal transduction pathway, and we also observed the high expression of many marker genes related to disease resistance in diseased samples, such as pathogenesis-related protein 1 (PR1).

3) Which ONE is the key pathogen for rood rot disease of Polygonatum? F. oxysporum, C. spaethianum and F. solani? And whether they have joint effects on the root rot disease?

Response: In the "Potential fungal pathogens and beneficial microbiota" section, we proved the potential pathogenic fungi and found potential biocontrol bacteria through culture and isolation experiments. In the above process, we inoculated the

top 4 most abundant fungi isolated from the diseased plants, such as *Fusarium oxysporum* (17.84%), *Candida sp.* (10.80%), *Colletotrichum spaethianum* (10.33%) and *Fusarium solani* (5.63%). However, only plantlets inoculated with *Fusarium oxysporum* and *Colletotrichum spaethianum* isolates developed disease phenotypes (Fig. 5B-b and c), and no obvious symptoms were observed after inoculation with *Candida sp.* or *Fusarium solani* isolates. At the same time, we also focused on bacteria with significant differences between healthy and diseased plants in the rhizomes and rhizosphere and identified potential biocontrol *Streptomyces* by choosing the different bacteria in the two samples. However, we did not conduct a coinoculation experiment, and we will consider using coinoculation settings of multiple pathogens and biocontrol bacteria in the next experiment.

4) As shown in Figure 1, the relative contents of rhizome saponins and other metabolites were lower in healthy plants when compared with diseased ones, how to explain it? Do those metabolites can induce plant pathogens?

Response: As explained in questions 1) and 2), transcriptome and metabolome results indicate that the physiological and metabolic processes of plants are altered due to the invasion of pathogens. Furthermore, *P. cyrtonea* quality markers (Q-Marker), such as polysaccharides (and glycosidase) (Buscaill et al., Science, 2019; Ding and Ding, Cell, 2020; Ding et al., 2018; Gallego-Giraldo et al., New Phytol, 2018; Gigli-Bisceglia et al., Cell host & microbes, 2020). and saponins (mainly including steroidal and triterpene saponins) (Orme et al., PNAS, 2019; Ribeiro et al., Plant Cell, 2020; Trda et al., 2019), are natural antimicrobial compounds (or compounds required for the biosynthesis of antimicrobial defense compounds) that increase the resistance of plants to pathogens, and they are important primary and secondary metabolites. We elaborate and discuss this part in discussion part: "Diseased rhizome transcription and metabolism may be involved in the plant defense response",

5) Microbiota assembly? Which kind of microbial assembly strategy in diseased and healthy plants, stochastic or deterministic, R or K strategy?

Response: I apologize that we did not specify the strategies of microbial assembly and the dynamics of community change, which may cause you to have questions about this section. In the study of the soil microbial ecology of model plants and other crops, the assembly strategy of plant microbiota after disease is a topic of concern. Previous studies found that there may be a deterministic process because several types of nutrient-using microorganisms dominate and have high abundance in the end. However, due to the lack of continuous dynamic sampling in our research, we could not draw a definite conclusion for the time being, which was also our concern when we drafted the title of the manuscript. We before selected three topics for the titles as follows "Multiomics reveals the effect of root rot on polygonati rhizome and identifies potential pathogens/identifies functional microbes/soil microbiota assembly". Finally, we changed the title of the manuscript to "Multiomics reveals the effect of root rot on polygonati rhizome and identifies potential pathogens", to avoid causing confusion to readers.

Other issues:

1) English should be improved by native speaker within the field. Please check the spelling mistakes and grammar throughout the whole manuscript.

2) Provide detailed information in Materials and Methods, for example Filed experiments description, and especially for the Bioinformatic analysis section.

3) Please carefully revise the discussion section.

Response: We sincerely thank you for the kind reminder. We have revised the above issues and improved the manuscript writing by working with a language editing service company to make it more relevant to the subject of the manuscript. In the first edition of the manuscript, due to the large amount of content in the methods section design, such as transcriptome, metabolome, microbiome and bioinformatics analyses, we included part of the content in Additional file 3 (Supplementary Methods 1-3).

January 23, 2022

Prof. Liu Guizhou
Xishuangbanna Tropical Botanical Garden, Chinese Academy of Science
CAS Key Laboratory of Tropical Plant Resources and Sustainable Use
Mengla
China

Re: Spectrum02385-21R1 (Multiomics reveals the effect of root rot on polygonati rhizome and identifies potential pathogens)

Dear Prof. Liu Guizhou:

Thank you for submitting your manuscript to Microbiology Spectrum. The comments of the reviewers are included at the bottom of this letter.

The reviewers have suggested publication in Microbiology Spectrum. Therefore, I invite you to respond to the reviewers' comments and revise your manuscript.

Link Not Available

Sincerely,

Yonglin Wang

Journals Department
Reviewer comments:

Reviewer #2 (Comments for the Author):

Current manuscript was much improved, but the revised title of "Multiomics reveals the effect of root rot on polygonati rhizome and identifies potential pathogens" is still not clearly highlight the goal of the study, please revise again.

In addition, the links between plant metabolite and microbiome is still missing, please provide the related results.

Staff Comments:

Preparing Revision Guidelines

Please return the manuscript within 60 days; if you cannot complete the modification within this time period, please contact me. If you do not wish to modify the manuscript and prefer to submit it to another journal, please notify me of your decision immediately so that the manuscript may be formally withdrawn from consideration by Microbiology Spectrum.

Dear Prof. Yonglin and Reviewers,

We sincerely thank you for your expert evaluation and specified comments on our manuscript, which will definitely help us to improve the quality of the manuscript. We have studied the comments carefully and revised the manuscript following your suggestions, which we hope will be met with approval. The main corrections in the paper and the responses to the reviewers' comments are as follows:

Reviewer #2: (Comments for the Author):

Current manuscript was much improved, but the revised title of "Multiomics reveals the effect of root rot on polygonati rhizome and identifies potential pathogens" is still not clearly highlight the goal of the study, please revise again.

Response: We sincerely thank the reviewer for this comment. We changed the title of the manuscript to "Multiomics reveals the effect of root rot on polygonati rhizome and identifies pathogens and biocontrol strain".

In addition, the links between plant metabolite and microbiome is still missing, please provide the related results.

Response: Thank you for asking this interesting question. In this study, we first focused on two metabolites (in rhizome metabolites, not root exudates): saponins and polysaccharides which metabolites are medicinal functional components of *Polygonatum*. Based on targeted metabolite results, we found that the rot disease caused the increase of these two compounds (but the plant still died and the yield decreased), and combined with the transcriptome results, we predicted that these may also be involved in the plant defense system as defense compounds. Studies have shown that plants can influence their microbiome by secreting various metabolites and, in turn, the microbiome may also impact the metabolome of the host plant. For the correlation between plant metabolites and microbiota, we have considered using traditional analysis (such as Pearson's r Correlation) and machine learning methods (Such as LDA) to predict the correlation between plant metabolites and microbiota. But this only hypothesis that based on the correlation is not reflect the actual plant metabolites (or root exudates) shape rhizome microbiota or rhizosphere/root microbiota promoting effect of plant metabolites (Luo et al., 2021. Enrichment of *Burkholderia* in the Rhizosphere by Autotoxic Ginsenosides to Alleviate Negative Plant-Soil Feedback. *Microbiology Spectrum*; Pang et al., 2021. Linking Plant Secondary Metabolites and Plant Microbiomes: A Review. *Front Plant Sci*; Sasse et al., 2018. Feed Your Friends: Do Plant Exudates Shape the Root Microbiome? *Trends Plant Sci*.). Which need to make clear the relation between plant metabolites and rhizome/rhizosphere microbiota should actual strains inoculation experiment, unfortunately, compared with the model plant *Arabidopsis thaliana*, chinese medine plants likes *Polygonatum cyrtoneuma* Hua and Sanqi [*Panax notoginseng* (Burk.) et al growth cycle is long, It takes at least two to three years to determine the changes in plant metabolites after inoculation (Xu et al., 2021, AutotoxinRg1 induces degradation of root cell walls and aggravates root rot by modifying the rhizospheric microbiome, *Microbiology Spectrum*). We hope you can understand that we will continue to pay attention to the relationship between plant functional metabolites/root exudates and rhizome/root/rhizosphere microbiota and pathogens in subsequent studies.

February 10, 2022

Prof. Liu Guizhou
Xishuangbanna Tropical Botanical Garden, Chinese Academy of Science
CAS Key Laboratory of Tropical Plant Resources and Sustainable Use
Mengla
China

Re: Spectrum02385-21R2 (Multiomics reveals the effect of root rot on polygonati rhizome and identifies pathogens and biocontrol strain)

Dear Prof. Liu Guizhou:

I am pleased to inform you that your paper should become acceptable for publication, and I am forwarding it to the ASM Journals Department for publication. You will be notified when your proofs are ready to be viewed.

Sincerely,

Yonglin Wang
Editor, Microbiology Spectrum
